# Surgical Site Infection in Cardiac Surgery

**DOI:** 10.3390/jcm11236991

**Published:** 2022-11-26

**Authors:** Agnieszka Zukowska, Maciej Zukowski

**Affiliations:** 1Department of Infection Control, Regional Hospital Stargard, 73-110 Stargard, Poland; 2Department of Anesthesiology, Intensive Care and Acute Intoxication, Pomeranian Medical University, 70-204 Szczecin, Poland

**Keywords:** SSI, DSWI, mediastinitis, cardiac surgery, prevention, treatment

## Abstract

Surgical site infections (SSIs) are one of the most significant complications in surgical patients and are strongly associated with poorer prognosis. Due to their aggressive character, cardiac surgical procedures carry a particular high risk of postoperative infection, with infection incidence rates ranging from a reported 3.5% and 26.8% in cardiac surgery patients. Given the specific nature of cardiac surgical procedures, sternal wound and graft harvesting site infections are the most common SSIs. Undoubtedly, DSWIs, including mediastinitis, in cardiac surgery patients remain a significant clinical problem as they are associated with increased hospital stay, substantial medical costs and high mortality, ranging from 3% to 20%. In SSI prevention, it is important to implement procedures reducing preoperative risk factors, such as: obesity, hypoalbuminemia, abnormal glucose levels, smoking and *S. aureus* carriage. For decolonisation of *S. aureus* carriers prior to cardiac surgery, it is recommended to administer nasal mupirocin, together with baths using chlorhexidine-based agents. Perioperative management also involves antibiotic prophylaxis, surgical site preparation, topical antibiotic administration and the maintenance of normal glucose levels. SSI treatment involves surgical intervention, NPWT application and antibiotic therapy

## 1. Introduction

Surgical site infections (SSIs) are one of the most significant complications in surgical patients and are strongly associated with poorer prognosis. SSIs affect up to 10–20% of patients undergoing major surgery [1]. In the USA, SSIs cause over 90,000 readmissions, extend the average length of hospital stay by up to 9.7 days and increase medical costs by $700 millions each year [2]. Due to their aggressive character, cardiac surgical procedures are associated with an even greater degree of postoperative infection risk. SSIs in cardiac surgery patients are one of the most problematic hospital acquired infections, due to the immediate proximity of the surgical site to vital organs.

## 2. Definitions

The Centres for Disease Control and Prevention (CDC) define a SSI as an infection that occurs within 30–90 days after a surgical procedure. Superficial incisional SSIs are defined as occurring within 30 days after surgery, whereas deep and organ/space SSIs are defined as occurring within 30–90 days after a procedure. Superficial incisional SSIs involve the skin and subcutaneous tissue of the incision, whereas deep incisional SSIs include the deep soft tissues of the incision, e.g., fascial and muscle layers. Organ/space SSIs are the most serious form of SSI. They involve any part of the body deeper than the fascial or muscle layers manipulated during surgery. There are two specific types of superficial incisional SSIs: primary superficial incisional SSI, i.e., identified in the primary incision in a patient that has undergone a surgical procedure with one or more incisions, and secondary superficial incisional SSI, i.e., identified in the secondary incision in a patient that has undergone a surgical procedure with more than one incision. A similar classification can be applied to deep incisional SSIs [3,4,5].

## 3. Characteristics of SSIs in Cardiac Surgery

Given the specific nature of cardiac surgical procedures, sternal wound and graft harvesting site infections are the most common SSIs in cardiac surgery patients. Sternal wound infections are classified into superficial and deep, depending on the structure involved and into early and late—depending on the timing of onset. Superficial sternal wound infections (SSWIs) involve the skin, subcutaneous tissue and/or pectoralis fascia, whereas deep sternal wound infections (DSWIs) involve tissues beneath the fascia [5]. Deep sternal wound infections include deep incisional infections and mediastinitis, associated with sternal osteomyelitis with or without retrosternal space involvement [6]. Mediastinitis is one of the most serious postoperative infections in cardiac surgery patients. In addition to the CDC criteria for mediastinitis [4], there are also clinician-developed diagnostic criteria for postoperative mediastinitis [7]. The diagnosis of postoperative mediastinitis must meet at least one of the following criteria: positive microbiological testing results from mediastinal tissue or fluid samples obtained during surgery or needle aspiration; evidence of mediastinitis during anatomic or histopathologic examination, the presence of at least one of the following symptoms with no other recognised cause, fever (>38 °C), chest pain or sternal instability and at least one of the following: purulent drainage from the mediastinum, evidence of an infective process in the mediastinum on an imaging examination, organisms cultured from blood or spontaneous purulent discharge from the mediastinal area. Importantly, mediastinitis is defined as an infection occurring within one year from surgery, regardless of whether or not an implant is in place [8,9]. In addition to those listed above, other less common clinical manifestations of SSI in cardiac surgery patients include; myocarditis, pericarditis, endocarditis, intraabdominal infection, infection of the lower respiratory tract and blood stream infection [3].

## 4. Clinical Symptoms of SSIs

The diagnosis of postoperative infection in cardiac surgery patients may be complicated due to the difficulty in distinguishing the symptoms of infection from a systemic inflammatory response caused by extensive tissue damage or the use of cardiopulmonary bypass during surgery [10]. The clinical symptoms of SSI in cardiac surgery patients vary depending on the type of infection. The most common manifestations of superficial SSIs after cardiac surgery are redness, exudate, subcutaneous fluid collections and wound dehiscence; the symptoms are always locally limited and the sternum is stable and non-tender on palpation with both hands. The signs of deep SSIs in cardiac surgery patients include, in addition to the symptoms listed above, sternal instability, chest pain, fever and the presence of a purulent discharge in the mediastinum [11,12,13]. Complications of sternotomy have a range of clinical manifestations, from sterile dehiscence to purulent mediastinitis with sternal osteomyelitis. It should be noted that only a surgeon can assess the depth of the infection and the nature of fluid collections during sternal revision surgery. The clinical course of mediastinitis ranges from subacute, where the overall condition of the patient is stable, to fulminant, where the patient is in a critical condition and requires immediate intervention. The first signs of mediastinitis in cardiac surgery patients usually occur between day 14 and 30 after surgery. Clinical symptoms of mediastinitis usually include tachycardia and fever. In more severe cases, sepsis and septic shock can develop and the patient may require intensive therapy. Systemic signs of sepsis are a strong indicator of mediastinal involvement [8,14].

## 5. Epidemiology

It is difficult to estimate the incidence of infection in cardiac surgery patients, with reported rates ranging between 3.5% and 26.8% [10,15,16,17,18]. SSI incidence after cardiac surgery varies depending on the quality of local epidemiological surveillance, SSI definition used, patient profile and the type of cardiac procedure [19]. According to the literature, it ranges from 0.5% to 7.8% [20,21,22,23]. The incidence of superficial SSI ranges from 0.5% to 8% [7,21], whereas deep infections ranges between 0.5% and 5.6%. The incidence rates of DSWI reported by most large cardiac surgery centres in Europe and the USA range between 1% and 2% per year [14,24]. Over the last 29 years, the incidence rate of mediastinitis has been decreasing, whereas the 30-day mortality rate has not changed. This is due, among other factors, to the more advanced age and poorer overall condition of patients undergoing cardiac surgery [25]. According to recent data, the incidence of post-sternotomy mediastinitis is 1–5%, with a rate >2% being an indicator of poor quality care in the surgery centre [14,26]. Undoubtedly, DSWIs, including mediastinitis, in cardiac surgery patients remain a significant clinical problem, as they are associated with an increased length of hospital stay, substantial medical costs and high mortality ranging from 3% to 20% [7,21,27,28,29,30] and can reach up to 50% [13,28,31].

## 6. Risk Factors

Several preoperative, intraoperative and postoperative risk factors for SSIs after cardiac surgery have been identified. Preoperative risk factors include, diabetes, obesity, advanced age, high logistic EuroSCORE (European System for Cardiac Operative Risk Evaluation logistic regression), COPD, heart failure with left ventricular dysfunction, smoking, female sex, breast size, renal failure, Staphylococcus aureus (*S. aureus*) carriage, peripheral vascular disease and prolonged length of hospital stay before surgery. Intraoperative risk factors are as follows: urgency/emergency of the surgical procedure, use of bilateral internal mammary arteries, combined surgical procedures, e.g., CABG + AVR (coronary artery bypass graft + aortic valve replacement), prolonged surgery duration, duration of cardiopulmonary bypass and aortic clamping. Postoperative risk factors include, postoperative respiratory failure, need for inotropic support and prolonged ICU stay [8,14,24,31,32,33,34,35]. *S. aureus* carriage, obesity, diabetes, COPD and renal failure are considered to be the most significant of all the risk factors listed above [13,24,32,36,37,38,39,40].

Numerous clinical studies have described obesity (i.e., BMI > 30 kg/m^2^) as an independent risk factor for infectious complications in cardiac surgery patients. As it is one of the modifiable risk factors, many cardiac surgery centres recommend that obese patients should, if possible, lose weight before surgery. However, there is no clear evidence to support delaying surgery until the patient has achieved sufficient weight loss. Nevertheless, when cardiac surgery is to be performed on an obese patient, it is necessary to adjust the doses of prophylactic antibiotics, thoroughly prepare the surgical site and reinforce the closure of the wound to prevent dehiscence. In addition, some authors recommend sternal closure using negative pressure wound therapy (NPWT) [13,26].

Preoperatively undiagnosed diabetes and postoperative hyperglycaemia are also associated with an increased risk of SSI in cardiac surgery patients. Therefore, preoperative screening for diabetes and hyperglycaemia is obligatory in most cardiac surgery centres. In patients with high preoperative HbA1c levels (>6.5–7%), the optimisation of glucose control prior to surgery is recommended to reduce the risk of mediastinitis. Moreover, it is known that appropriate perioperative glucose control with insulin infusion is necessary, as glucose levels >200 mg/dL in the first two postoperative days may be associated with an increased rate of sternal infection [33,36,41,42,43]. In patients with hyperglycemia who require urgent surgery, perioperative intravenous insulin infusion is the most effective form of rapid glucose control. Most cardiac and thoracic surgery associations and societies, including the Society of Thoracic Surgeons (STS), the American Association for Thoracic Surgery (AATS) and the European Association for Cardio-Thoracic Surgery (EACTS), recommend that perioperative glucose levels in patients undergoing cardiac surgery should be maintained at <180 mg/dL [44,45,46], whereas guidelines by the Spanish Society of Thoracic and Cardiovascular Surgery (SECTCV) and the Spanish Society of Cardiovascular Infections (SEICAV) recommend maintaining intraoperative glucose levels at 110–180 mg/dL [26]. 

Due to the heterogeneity of cardiac surgical procedures, and thus differences in invasiveness, cardiac operations have been classified into different SSI risk categories. Heart transplant is categorised as carrying the highest risk of SSI. It is followed by coronary artery bypass graft (CABG) with peripheral grafts used, CABG only with sternal incision and other types of cardiac surgery [3]. The lowest rate of local infectious complications is reported for isolated valve surgery [47].

There are a number of scales which can be used to assess and stratify the risk of SSI after cardiac surgery. In clinical practice, regardless of the risk assessment method used (the Brompton Harefield Infection Score (BHIS), the Australian Clinical Risk Index (ACRI), the Infection Risk Index in Cardiac surgery (IRIC), the Gatti score, the Northern New England Cardiovascular Disease Study Group prediction rule for mediastinitis, the Friedman score, the Alfred Hospital risk index A), the patient should be classified into a low-, medium- or high-risk category for developing SSI [48,49,50,51,52,53].

## 7. Pathomechanism

Systemic inflammatory response syndrome is usually initiated within a couple of hours following surgical injury. This acute non-specific reaction is an inflammatory response to both tissue damage and blood loss and is initiated by endogenous cytokines released at massive rates from the damaged tissue [54,55]. It has been shown that most dangerous associated molecular patterns (DAMPs) and alarmins can be mobilised into circulation from the injured cells or tissues by surgical insult [56]. The inflammatory response is initially beneficial since it helps to remove tissue debris. However, if it is not balanced by homeostatic anti-inflammatory mechanisms, it undermines the integrity and repair of tissue and may result in a depressed immune response due to the extensive death of immune effector cells [57]. The increased non-specific inflammatory response at the early stage of surgical injury is usually accompanied by the suppression of the surgical patient’s ability to initiate an effective defence against pathogens. In addition, anaesthetic management may have an impact on the effectiveness of the immune response in surgical patients. It has been demonstrated that high doses of opioids, such as remifentanil, may induce immunosuppression through the activation of leucocyte-expressed opioid receptors and may increase a susceptibility to infection resulting from opioid withdrawal in the postoperative period [58].

Contamination during surgery is considered to be the main cause of SSI in surgical patients. Cardiac surgeries are classified as “clean” procedures, as they do not involve the opening of any contaminated space (e.g., the intestines, airways or urinary tract) [5]. However, despite the use of modern surgical techniques, the wound is still colonised by endogenous and exogenous bacteria in almost every patient during cardiac surgery [59]. In addition, the use of surgical materials, such as sutures, synthetic materials and haemostatic sponges, increase the risk of wound contamination [60]. The most common source of SSI is the patient’s skin microbiome, with *S. aureus* being predominantly isolated from infected wounds. It has been shown that approximately 80% of *S. aureus* surgical site infections and bacteraemia are caused by the patient’s own bacteria [26,61,62]. Comparative analyses of bacterial DNA have demonstrated that the genotype of *S. aureus* isolated from the sternum of patients with mediastinitis and that of *S. aureus* isolated from the patient’s nares are identical [7].

Infections with flora originating from the surgical team have become rare since the introduction of strict aseptic techniques, and if they do occur, they usually result from ineffective infection surveillance in hospital [13]. As wound contamination mainly occurs during surgery and during the early stage of wound healing, most cases of DSWI are seen in the first three weeks post-surgery. After this initial period resternotomy, wound dehiscence, pericardiocentesis, percutaneous electrode placement or sepsis may also be caused DSWI [24].

## 8. Aetiology

A number of studies describe the aetiology of SSI after cardiac surgery. Most authors agree that almost two-thirds of the microorganisms isolated from infected patients are Gram-positive bacteria (60–80%), including *S. aureus*, coagulase-negative *Staphylococcus* and, in fewer cases, *Propionibacterium acnes* (currently: *Cutibacterium acnes*) [13,26,27,63]. *Staphylococci* are the main bacteria responsible for postoperative SSIs, even though the reported number of SSI cases caused by this pathogen varies. *S. aureus* accounts for 40–60% of the strains causing mediastinitis, and coagulase-negative *Staphylococci* are involved in 20–30% of mediastinitis cases [33]. 

*Staphylococci* are invasive pathogens responsible for a number of different infections, from mild, superficial skin infections to life-threatening bacteraemia with concomitant infective endocarditis associated with high mortality. They are commonly present on human skin and have numerous virulence factors. [64]. Gram-negative bacteria are less commonly isolated from patients after cardiac surgery (20–40%) and mainly include *Enterobacterales*, such as *Enterobacter* spp., *Escherichia coli*, *Klebsiella pneumoniae*, *Proteus mirabilis*, *Serratia* spp. and *Citrobacter* spp. Non-fermenting bacteria, such as *Pseudomonas* spp. and *Acinetobacter* spp., are significantly less commonly cultured in those patients [14,27,65,66]. Infections caused by Gram-negative bacteria mainly develop as a result of bacterial translocation from other sources of infection, develop earlier after surgery, than infections with Gram-positive bacteria and are associated with higher mortality [14,65]. Mediastinitis is rarely caused by *Candida* spp. (2–5%). Between 20% and 44% of cases of postoperative mediastinitis have a mixed aetiology [13,14,67].

## 9. Diagnostics

### 9.1. Microbiological Diagnostics

SSI diagnosis in cardiac surgery is based on clinical symptoms, X-ray imaging and microbiological diagnostics. One of the diagnostic criteria for SSI according to the CDC is microbiological criterion. However, the CDC definitions do not offer any detailed requirements to the methods used in microbiological diagnostics or the interpretation of bacteriological results [3].

Collecting samples for microbiological testing is key for establishing the aetiology of infection. With superficial infections, it is generally recommended to perform wound swab after surgical preparation. For deep infections, especially mediastinitis, it is recommended to perform a CT-guided aspiration biopsy if the sample has to be collected retrosternally with no direct access [26]. Retrosternal aspiration may also be useful in patients with suspected mediastinitis and postoperative sepsis with no local symptoms of infection (inflammation, exudate and/or sternal instability) and when there are no other possibilities to confirm the diagnosis [14,68]. Cultures from superficial wounds, fistulas or drain fluid should be interpreted with caution, as they do not always lead to the determination of a etiological factor, but often only indicate that a given region has been colonised. 

Most publications on microbiology of post-sternotomy mediastinitis present the results of blood cultures, wound swabs, drain and surgical samples and analyse correlations between the different sample types [69,70]. It is assumed that the multiple isolation of the same microorganism from superficial wounds or fistulas, especially in the case of *S.aureus* or Gram-negative bacilli, has a high predictive value as aetiological factors of mediastinitis [71]. Positive blood cultures in patients suspected of mediastinitis may be helpful in establishing aetiology. The presence of bacteraemia without other sources of infection within 90 days post-surgery may indicate mediastinitis, especially when *S.aureus* is isolated [72]. In such cases, the interpretation of positive cultures other than blood cultures is difficult and should be approached individually. Thus, the potential significance of the isolate as an aetiological factor of mediastinitis will depend on the pathogen, microbiological sampling site and clinical signs [12]. Negative cultures from the surgical site are not necessary before wound closure [73].

The routine use of non-culture-based methods for confirming the aetiology of mediastinitis is not recommended. Molecular methods may be considered in patients with mediastinitis and negative cultures or in patients diagnosed during antibiotic therapy [74]. If the establishment of aetiology with standard cultures is not possible, rare pathogen detection methods should be considered: specific serological tests (*Brucella* spp., *Coxiella* spp. and *Bartonella* spp.), and PCR and cultures in special, growth media (*Mycoplasma* spp., *Ureaplasma* spp., *Legionella* spp., *Nocardia* spp., fungi and mycobacteria) [69,75].

### 9.2. Diagnostic Imaging

Diagnostic imaging is often crucial to confirm the clinical diagnosis of a deep SSI, including mediastinitis. X-ray imaging has limited use, as it is difficult to distinguish whether the mediastinal widening is caused by postoperative haemorrhage, oedema or infection [76,77]. Computed tomography (CT) offers the highest diagnostic value in cardiac surgery patients with SSI of available radiology techniques, and is also a first-choice imaging technique when postoperative mediastinitis is suspected. It is recommended to perform CT in the 2nd week post-surgery, when gas or fluid collections are not normally present in the mediastinum. It is highly sensitive, but its specificity reaches 100% only after 14 days post-surgery. Furthermore, tomography can be considered for diagnostics in patients with fever and leucocytosis without DSWI symptoms and in those diagnosed with wound infection to determine the extensive nature of the infection [14,78,79]. CT is always recommended when mediastinitis is suspected, even if the diagnosis is certain, due to better treatment planning [26]. The routine use of magnetic resonance imaging (MRI) in cardiac surgery is not recommended, due to the strong artefacts caused by wires used for sternal closure [80]. Nuclear medicine imaging with the labelled leukocytes can also be used for diagnosing post-sternotomy mediastinitis and osteomyelitis [81]. Likewise, positron emission tomography (PET) is also considered a useful tool for the diagnosis and observation of cardiovascular infections [82].

## 10. SSI Prevention

SSI prophylaxis in surgical medicine involves a wide range of preventive pre-, intra- and postoperative measures [83,84]. Due to the specificity of cardiac surgeries, the standard perioperative management protocol includes some elements that are characteristic of these types of procedures, such as: preoperative optimisation of patient condition, preoperative screening and *S. aureus* decolonisation, surgical site preparation, antibiotic prophylaxis, topical antibiotic therapy and negative pressure wound therapy. The preventive measures for SSI in cardiac surgery has been presented in Table 1.

### 10.1. Preoperative Optimisation of Patient Condition

Poor preoperative nutritional status, especially hypoalbuminemia, is a risk factor of sternal wound infections following cardiac surgery. In patients with hypoalbuminemia <2.5 g/mL or total body weight loss >10% within 6 months before surgery, enteral nutritional support should be initiated 7–10 days prior to elective procedure [35,45]. The significance of body weight control in obese patients has been discussed in Section 6. It is recommended to resolve all infections involving systems and organs outside the cardiovascular system, such as urinary tract infection, pneumonia, intra-abdominal infection or skin and soft tissue infections before elective cardiac surgery [45]. 

Glycaemic control recommendation was discussed in Section 6.

Smokers who qualify for cardiac surgery should cease smoking at least 30 days before the procedure. Pulmonary physiotherapy and toilet should be performed to reduce the risk of postoperative infection and sternal dehiscence [26,45].

### 10.2. Staphylococcus aureus Decolonisation

*Staphylococcus aureus* is one of the leading aetiological factors in hospital-acquired infections and the main cause of SSIs and is also the case in cardiac surgery. *S. aureus* colonises human skin and mucous membranes and in the human population about 50% of people carry *S. aureus*, of whom 20% are persistent carriers and another 30% are short-term hosts [62]. It is estimated that 1–1.5% of the population are carriers of methicillin resistant *Staphylococcus aureus* (MRSA), with MRSA carriage being more prevalent among patients who are repeatedly hospitalised, the elderly, diabetics and patients with immunodeficiency [8,85,86,87]. *S. aureus* is most frequently found in the nares [8,62,88,89]. Therefore, most societies recommend nasal swab testing to determine *S. aureus* carriage, while swabs from other sites (armpits, groins, rectum, etc.) are not recommended [26]. *S. aureus* infections are associated with a severe clinical course and numerous complications, especially when caused by MRSA strains. Currently, *S. aureus* carriage is believed to be a well-defined risk factor of SSI following cardiac surgery [33]. Thus, it seems necessary to take all possible actions towards reducing this risk, including eradication before elective surgery [90]. 

Most data concur that *S. aureus* decolonisation before elective cardiac surgery significantly reduces the postoperative incidence of SSIs [24,32,34,85,91,92,93,94,95]. In contrast, the authors of a 2017 Cochrane meta-analysis were not able to demonstrate any potential benefit of *S. aureus* decolonisation in SSI prevention of cardiac surgery patients, though only two studies met the inclusion criteria for the analysis and one was very small and poorly described [96]. Likewise, in a 2020 meta-analysis, Tang et al. demonstrated the effectiveness of decolonisation in reducing the SSI rate, but this did not apply to cardiac surgery patients as no statistical significance was observed in that group [97]. This most likely resulted from the heterogeneity of the groups in included studies. In turn, a recent systematic review and meta-analysis based on 17 studies, including a total of over 26,000 post-sternotomy patients, revealed a significant SSI reduction in a group undergoing decolonisation [98]. Despite partially contradictory results of studies regarding the effectiveness of *S. aureus* eradication in SSI prevention, most scientific societies clearly recommend this procedure [7,8,26,66,83,99]. 

The generally accepted management protocol prior to cardiac surgery involves determining *S. aureus* carriage within about 15–30 days before the procedure. In the case of a positive result, carriers are administered intranasal mupirocin ointment and chlorhexidine baths [26,83,90,100]. Mupirocin is applied 2–3 times a day into both nostrils for 4–7 days (usually 5 days). At the same time, daily baths with chlorhexidine-based agents are performed. Many societies recommend different protocol durations: the WHO recommends following a protocol for 5 days, without providing the exact moment of initiation [83], the National Institute for Health and Care Excellence (NICE) recommends using it from 2 days before surgery to 3 days post [90], the Society of Thoracic Surgeons (STS) recommends initiating the protocol at least one day before surgery and continuing it to 2–5 days after [99] and finally, the SEICAV/SECTCV in their newest recommendations do not provide any exact details of protocol duration [26]. In patients requiring urgent surgery, without carriage assessment, mupirocin should be initiated as soon as possible and continued for 5 days post procedure [83].

Mupirocin is a drug with clearly established effectiveness in rapidly eradicating *S. aureus* from the nose and its efficiency has been confirmed in several randomised studies [26]. Unfortunately, the increasing resistance of *S. aureus* to mupirocin questions the effectiveness of a preoperative decolonisation of the patient. The action mechanism of Mupirocin involves isoleucyl-tRNA synthetase (IleRS) inhibition, and thus subsequently inhibits bacterial protein synthesis. Resistance to mupirocin is associated with the *ileRS* gene in bacteria. In the case of low-level resistance, point mutation occurs in the native *ileRS* gene, with MIC for mupirocin being 8–64 mg/L. High-level mupirocin resistance, in turn, is mediated by the plasmid-based *mupA* gene (so called *ileS-2*), which encodes for the alternate native *ileRS* gene. In this case, MIC values exceed 256 mg/L [62]. Even though *mupA* is strongly associated with high-level mupirocin resistance, its presence is also observed in strains with low-level resistance. The clinical significance of low-level mupirocin resistance remains unclear and *S. aureus* with intermediate MIC values is rare. The prevalence of mupirocin resistance among MRSA strains ranged from 0% in neonatal ICU patients, to 9.4% in patients staying in nursing homes, with considerably lower rates (0.3–1.2%) observed among MSSA strains [101]. 

Resistance to mupirocin is associated with high decolonisation failure rates. Susceptible MRSA strains are eradicated in 90% of cases following the administration of mupirocin. Successful decolonisation rate in patients with low-level mupirocin-resistant MRSA is only 29%. As for high-level mupirocin-resistant MRSA carriers, the success rate is 24% [62]. Apart from mupirocin, other agents used for *S. aureus* carriage eradication include neomycin, fusidic acid, chlorhexidine and octenidine. Compared to antibiotics, antiseptics are often active against a broad spectrum of microorganisms and can also eradicate other potential pathogens. They usually do not cause tissue damage and thus can be used on intact skin and some types of open wounds [102,103]. 

Currently, there are few studies comparing the effectiveness of mupirocin to other antibiotics and antiseptic agents, and most of these included orthopaedic patients. In a recent study conducted among patients undergoing hip and knee replacements (over 15,000 patients), 20% were found to be MSSA carriers and received mupirocin, neomycin and octenidine for decolonisation. Mupirocin and neomycin showed similar effectiveness at decolonisation (90%), whereas nasal octenidine gel was found to be significantly less effective (50%) [104]. The search for mupirocin alternatives yielded new possibilities for *S. aureus* eradication, including lysostaphin, 70% ethanol combined with emollients, omiganan pentahydrochloride, tea tree (*Melaleuca alternifolia*) oil, natural honey, probiotics and specific bacteriophages. Currently there are no studies comparing the effectiveness of these substances with mupirocin in clinical settings [62,105].

### 10.3. Preoperative Bathing 

One of the key interventions preparing the patient for surgery is a preoperative bath with soap or an antibacterial agent on the evening before surgery and/or in the morning immediately prior to surgery [83,90,106]. Since no advantage of washing with antibacterial agents compared to regular soap has been demonstrated [107,108], the literature on cardiac surgery does not recommend the use of the antibacterials, but nevertheless points to the benefits of using chlorhexidine-based agents [7,8,18,100]. Furthermore, the SEICAV/SECTCV note that the studies which did not demonstrate the superiority of chlorhexidine over other agents only included a small group of cardiac surgery patients [26].

### 10.4. Surgical Site Preparation

Preoperative hair removal from the surgical site is not obligatory. However, if necessary, it is recommended to use clippers with a single-use blade immediately before surgery (not the day before). Shaving is strongly discouraged, as it increase the risk of micro-injuries and bacterial growth on the patient’s skin [26,35,83,90,100].

For antiseptic skin preparation immediately before surgery, the WHO strongly recommends the use of alcohol-based chlorhexidine solutions [83]. If chlorhexidine is contraindicated, an alcohol-based solution of povidone-iodine is recommended [90]. In cardiac surgery, alcohol-based chlorhexidine solutions are the antiseptics of choice [8,26,35,100].

### 10.5. Perioperative Antibiotic Prophylaxis—PAP

Perioperative antibiotic prophylaxis (PAP) is one of the most important measures for the prevention of surgical site infections (SSIs) in cardiac surgery. The most commonly used antibiotic in prophylactic protocol is cefazolin, administered intravenously as bolus (2–3 g based on patient weight, 2 g < 120 kg, 3 g > 120 kg) within 0–60 min before incision, optimally within 15–30 min before incision [45,66,83,99,109,110,111,112]. When comparing the effectiveness of first-, second- and third-generation cephalosporins, it has been demonstrated that the administration of cefazolin as prophylaxis for infection prevention in cardiac surgery was associated with the lowest rate of SSIs. Importantly, when compared to cefazolin, the use of third-generation cephalosporins for PAP is associated with a 3-fold increase in post-operative infection incidence, longer hospital stays and increased treatment costs [15,63]. In patients at high risk for MRSA infection (colonised patients/patients infected with MRSA; history of MRSA colonisation/infection; stay at a hospital/facility/nursing home with a high prevalence of MRSA infections), vancomycin should be additionally administered [7,45,113]. Vancomycin dosage is 1–1.5 g i.v. and depends on patient body weight, with 1.5 g to be used in patients >80 kg. Vancomycin should be administered within 15/30–120 min before incision. It is recommended that the infusion rate for this drug should be 1 g/1 h and 1.5 g/1.5 h [46,66,110,114]. 

Patients at high risk for penicillin/cephalosporin/beta-lactam allergy (history of: anaphylaxis, angioedema, bronchospasm, urticaria, toxic epidermal necrolysis (TEN), drug reaction with eosinophilia and systemic symptoms (DRESS), Stevens-Johnson syndrome) should be given vancomycin for PAP, 1–1.5 g i.v. based on the patient weight. Patients wih an intolerance to vancomycin should receive clindamycin 900 mg i.v. Apart from receiving prophylactic vancomycin or clindamycin, allergic patients should also be given an antibiotic covering Gram-negative bacteria, e.g., a single dose of gentamicin 5 mg/kg i.v. (there is no need to continue its administration postoperatively). If surgery lasts longer than 4 h from the administration of the first dose of cefazolin, the drug should be re-dosed. In case of intraoperative massive blood loss (>1500 mL) requiring fluid resuscitation, antibiotics used for PAP should be re-administered [7,8,45,66,99,110,111].

Another important consideration in cardiac surgery is cardiopulmonary bypass (CPB), which has a significant effect on the pharmacokinetic parameters of drugs (protein binding, changes in the distribution volume, changes related to haemodilution and drug deposition in the CPB system). These changes also relate to antibiotics used for PAP, in which the standard dosing does not take the aforementioned modifications into account. Consequently, personalised PAP seems to be an optimum approach which however, requires further studies. Antibiotics with a short half-life, e.g., cefazolin or cefuroxime, should certainly be re-dosed at 4 h following the administration of the first dose. However, the available literature indicates that in many patients cefazolin levels fall below the therapeutic range after 120 min. Subsequent doses of gentamicin are not recommended when using CPB [7,115,116].

Generally, antibiotic prophylaxis in cardiac surgery is applied for the first 24–48 h from the administration of the first dose of the drug [117]. Recent recommendations do not advise prolonging PAP beyond 24 h, even if chest drainage is placed [46,66,106,110,118,119]. It is currently recognised that prolonged PAP, more than 24 h, does not reduce the risk of SSI, but it increases the risk of adverse effects of antibiotics, mainly associated with a disturbed function of the gastrointestinal tract (including *Clostridioides difficile* infections), kidneys, hematopoietic system and liver [10,17,113,118,120,121]. Perioperative antibiotic prophylaxis (PAP) in cardiac surgery has been presented in Table 2.

### 10.6. Topical Antibiotic Administration

Reports on the topical administration of antibiotics in the form of ointments, powders, solutions, aerosols, etc., containing vancomycin, cefazolin, gentamicin, neomycin, ampicillin, cephaloridine, rifampicin bacitracin, etc., in cardiac surgery are contradictory and the evidence for the effectiveness and safety of such a procedure is limited and insufficient [26,90,122,123,124,125,126]. Due to the risk of inducing hypersensitivity to the substances used and particularly concerns about increasing antimicrobial resistance associated with such practices, it is currently recommended to not use topical antibiotics [84,90]. Many guidelines, including those issued in 2021, make a direct recommendation to not use topical antibiotics before wound closure in cardiac surgery patients [26,66,106,110]. Only the AATS recommends applying topical antibiotics to the sternal edges upon opening and prior to closing [45].

#### Gentamicin-Impregnated Collagen Sponges

One form of topical antibiotic administration is the use of gentamicin-impregnated collagen sponges (GICS). Gentamicin is an aminoglycoside antibiotic with a broad spectrum of antibacterial activity, mainly against Gram-negative bacteria, with lower activity against Gram-positive organisms. Gentamicin is also highly active against multidrug-resistant strains. In both Gram-positive and multidrug-resistant bacterial infections, gentamicin is usually combined with beta-lactam antibiotics to obtain better treatment outcomes using a synergy effect. The bactericidal effect of gentamicin is strictly related to its serum concentration. When used systemically, gentamicin is associated with a number of adverse effects (mainly nephrotoxicity and ototoxicity occurring when the drug concentration exceeds 10–12 mg/L), therefore they are used topically in cardiac surgery to minimise the risk of complications in burdened patients. [127,128].

Currently, GICSs are produced from biodegradable collagen, which is a haemostatic agent and whose polymer structure makes it a perfect carrier for gentamicin. Collagen is completely reabsorbed, with the reabsorption time depending on the conditions at the implantation site. GICS application results in very high gentamicin concentrations in the sternal region and in the pericardial fluid (about 300 mg/L), considerably exceeding therapeutic levels (4–10 mg/L), and low serum concentration of the antibiotic (up to 1 mg/L), which makes it topically effective even against gentamicin-resistant strains and, at the same time, safe for the patient. In patients with renal failure, GICS must be used with caution. High gentamicin levels may persist for 3–4 days following GICS implantation. At present, there are various techniques for GICS application, with the insertion region playing an important role, as it is associated with different gentamicin distribution in tissues. GICSs implanted posteriorly to the sternum mainly prevent deep sternal wound infections (DSWI), whereas those placed anteriorly are more effective in preventing superficial sternal wound infections (SSWI) [23,67]. Gentamycin sulphate, which is used in GICS, is a water-soluble antibiotic, therefore sponge soaking prior to its insertion may cause considerable drug loss, which may be clinically significant and reduce GICS effectiveness [23,67,129]. Concerns about inducing antibiotic resistance during topical drug administration also apply to GICS. Pharmacokinetic studies of GICS have demonstrated a high topical concentration of the drug immediately after implantation, its low serum concentration and rapid elimination, which significantly reduces the risk of gentamicin-resistant strain selection.

In contrast to antibiotics applied as ointments, solutions, aerosols or gels, etc., the use of GICS in cardiac surgery patients appears to be more beneficial. Numerous studies indicate that GICSs significantly reduce the incidence of SSIs, both deep and superficial in cardiac surgery patients [23,129,130,131,132,133,134,135]. Due to the use of different protocols of managing cardiac surgery patients, the 2020 NICE recommendations allow practitioners to consider using gentamicin-collagen sponges in cardiac surgery, emphasising that the evidence for their effectiveness is limited [90]. Despite numerous publications demonstrating the benefits of implanting GICS in patients at high risk of SSI, there is not wide consensus within scientific societies on the use of GICS in cardiac surgery due to the lack of randomised trials involving homogenous patient groups.

### 10.7. Negative Pressure Wound Therapy

An important method of managing postoperative wounds in cardiac surgery is negative pressure wound therapy (NPWT), used both as prevention and treatment. The prophylactic use of NPWT, e.g., Prevena (KCI) or PICO (Smith and Nephew), helps to hold wound edges together to prevent dehiscence, decreases lateral tension and oedema, increases tissue perfusion, stimulates granulation and reduces bacterial colonisation by isolating the wound from potential sources of contamination [31,136,137]. Prospective non-randomised studies comparing the elective use of NPWT with standard wound closure techniques in a group of patients at high risk of mediastinitis undergoing open heart surgery demonstrated a significant reduction in postoperative SSI incidence [138,139,140]. Despite the lack of randomised controlled studies, some societies recommend the prophylactic use of NPWT in cardiac surgery patients at high risk of mediastinitis [26,84]. 

NPWT, which is now commonly used in wound management, has become a significant therapeutic option among cardiac surgery patients with postoperative SSI. Apart from the previously mentioned increase in tissue perfusion and stimulation of granulation, NPWT enables a continuous removal of tissue debris and excess fluid from the wound [35]. It has been demonstrated that NPWT provides better treatment outcomes than standard management, offers lower infection recurrence rates, shorter ICU and hospital stay [7,141,142,143]. Furthermore, some studies have shown a significant mortality decrease in patients treated with NPWT [3]. Consequently, most societies recommend the use of NPWT whenever possible in patients with post-sternotomy deep wound infections [8,26,45]. It is important that the NPWT dressing should not be maintained for more than 7 days in patients who require mediastinal lavage during infection treatment. In addition, negative drain fluid cultures should not be used as criteria for discontinuing or continuing drainage [144].

## 11. SSI Therapy in Cardiac Surgery

The standard of care for post-sternotomy wound infections involves surgical treatment and antibiotic therapy. Therapeutic management should be personalised depending on the clinical condition of the patient, the extent of the infection and microbiological results. The surgical treatment of superficial infections involves incision and drainage at the surgical site. In deep infections, surgical treatment is key and must be adjusted to a clinical situation. The surgical approach includes removing necrotic tissues, draining infected spaces and using techniques for sternum closure. In both situations, negative pressure wound therapy (described above) may be helpful. Obviously, the use of antibiotics in the therapy of SSI in cardiac surgery is a significant determinant of treatment success [45,143,145]. 

### Antibiotic Therapy SSIs in Cardiac Surgery

At the time of SSI diagnosis, a decision should be made whether to use empirical or targeted therapy, which offers greater treatment effectiveness. This decision depends on the patients clinical condition. Patients in severe condition require the immediate initiation of intravenous antibiotic therapy, in the case of stable patients, treatment should be determined by pathogen antibiotic susceptibility [26]. Empiric antibiotic therapy should cover potential pathogens, therefore it is important to know the current, local microbiological situation. In centres with low risk of MRSA infections, therapy may be initiated using piperacillin with tazobactam (if the Gram-negative bacilli with ESBL and/or AmpC resistance phenotype were not cultured) or alternatively carbapenem. In centres with a high risk of MRSA infections and patients in critical condition, antibiotic therapy should cover *S.aureus* MRSA and Gram-negative bacteria (drug selection based on local antibiotic susceptibility). One of the most important DSWI etiological factors are coagulase-negative *Staphylococci*, which are mostly resistant to methicillin (MR-CNS). Vancomycin, daptomycin or teicoplanin are most often used in the therapy of methicillin-resistant *Staphylococci* (MRSA and MR-CNS) [11,26,31,35]. Due to a low rate of fungal SSI in cardiac surgery, antimycotics should not be routinely used in empiric therapy. The use of antifungal agents may be considered in critically ill patients with risk factors for invasive fungemia. Therapeutic options for the treatment of *Enterococcal* mediastinitis, especially in patients with bacteraemia, have been adopted from guidelines for the management of endocarditis [26].

All microbiological test results should be consulted with an infectious disease specialist to optimize therapy. The duration of SSI therapy in cardiac surgery has not yet been clearly defined, superficial infections should be treated for 3–4 weeks, with intravenous therapy administered for the first 10–14 days. With deep infections, treatment duration depends on numerous factors (e.g., surgical intervention used, disease severity or the isolated pathogen)—the usual recommendation is 4–6 weeks of antibiotic therapy, with minimum 2–3 weeks intravenous therapy. For subsequent oral treatment, it is important to select an agent with high bioavailability. Some type of SSI, e.g., with bone involvement or infections with fungal aetiology, treatment can even be prolonged for several months [11,26,31,35,146,147,148] Antibiotic therapy options for surgical site infection in cardiac surgery is presented in Table 3.

## 12. Summary

Surgical site infections in cardiac surgery remain a significant medical problem due to the deteriorated postoperative quality of life, increased mortality, longer hospital stay and increased treatment costs. In SSI prevention, it is important to implement procedures which reduce preoperative risk factors, such as: obesity, hypoalbuminemia, abnormal glucose levels, smoking and *S. aureus* carriage. For the decolonisation of *S. aureus* carriers prior to cardiac surgery, nasal mupirocin administration together with baths using chlorhexidine-based agents is recommended. Perioperative management also involves antibiotic prophylaxis, surgical site preparation, topical antibiotic administration and the maintenance of normal glucose levels. SSI treatment involves surgical intervention, NPWT and antibiotic therapy.

## Figures and Tables

**Table 1 jcm-11-06991-t001:** The preventive measures for SSI in cardiac surgery.

Preventive Measures for SSI in Cardiac Surgery
Preoperative optimisation of the patient’s condition - nutritional support in patients with hypoalbuminaemia <2.5 g/mL or total body weight loss >10% within 6 months before surgery - body weight control in obese patients (i.e., BMI > 30 kg/m^2^) - screening for diabetes and hyperglycaemia - removal other sources of infection - smoking cessation - pulmonary physiotherapy (smokers, COPD patients) *Staphylococcus aureus* decolonisation - intranasal mupirocin ointment - chlorhexidine baths Preoperative bathing Surgical site preparation - hair removal (if necessary) with a clipper - skin preparation (alcohol-based chlorhexidine solutions) Perioperative antibiotic prophylaxis—PAP (tab 2) Perioperative glycemic control Appropriate surgical techniques Negative pressure wound therapy

**Table 2 jcm-11-06991-t002:** Perioperative antibiotic prophylaxis (PAP) in cardiac surgery.

Cardiac Procedures	Recommended Prophylaxis	Patients at High Risk of MRSA Infection	Patients at High Risk Penicillin/Cephalosporin/Beta-Lactam Allergy
CABG Cardiac Valve Surgery Aortic surgery	cefazolin 2–3 g IV based on the patient’s weight: 2 g < 120 kg 3 g > 120 kg	cefazolin 2–3 g IV based on the patient’s weight: 2 g < 120 kg 3 g > 120 kg PLUS vancomycin 1–1.5 g IV based on the patient’s weight: 1 g < 80 kg (1 h infusion) 1.5 g > 80 kg (1.5 h infusion) Patients at high risk of MRSA infection: - MRSA colonised patients - patients infected with MRSA - history of MRSA colonisation/infection - stay at a hospital/facility/nursing home with a high prevalence of MRSA infections	vancomycin 1–1.5 g IV based on the patient’s weight: 1 g < 80 kg (1 h infusion) 1.5 g > 80 kg (1.5 h infusion) OR clindamycin 900 mg IV for patients intolerant vancomycin PLUS gentamicin 5 mg/kg IV (single dose) BMI > 30 kg/m^2^: use adjusted body weightdo not continue postoperatively Patients at high risk penicillin/cephalosporin/beta-lactam allergy—history of: - anaphylaxis - angioedema - bronchospasm - urticaria - toxic epidermal necrolysis (TEN) - drug reaction with eosinophilia and systemic symptoms (DRESS) - Stevens-Johnson syndrome

CABG—Coronary Artery Bypass Graft; MRSA—methicillin-resistant *Staphylococcus aureus*.

**Table 3 jcm-11-06991-t003:** Antibiotic therapy options for surgical site infection in cardiac surgery.

Empiric Antibiotic Therapy		
Vancomycin * 15–20 mg/kg based on actual body weight q8–12 h i.v. or daptomycin 8–10 mg/kg/d i.v. + piperacillin/tazobactam 4.5 g q6h i.v. or meropenem 1 g q8h i.v.		
**Targeted Antibiotic Therapy**		
**Aetiology**	**First-Line Treatment**	**Alternative Treatment**
*Staphylococcus aureus* MSSA	- cloxacillin 2 g q6h i.v.	- cefazolin 2 g q8h i.v. - in the case of non-immediate reaction: - cefazolin 2 g q8h i.v. in the case of immediate reaction: - vancomycin * 15–20 mg/kg based on actual body weight q8–12h i.v.
*Staphylococcus aureus* MRSA	- vancomycin * 15–20 mg/kg based on actual body weight q8–12h i.v. or - daptomycin 8–10 mg/kg/d i.v.	- teicoplanin: three loading doses of 400 mg i.v. administered q12h followed by 400 mg/d i.v. - ceftaroline 600 mg q12h i.v.
*Streptococcus* spp.	- benzylpenicillin 5–6 MIU q6h i.v.	- ampicillin 2 g q4–6h i.v. - ceftriaxone 2 g q24h i.v. in the case of immediate reaction: - vancomycin * 15–20 mg/kg based on actual body weight q8–12h i.v.
*Enterococcus faecalis ****	HLAR (−) strains - ampicillin 2 g q4–6h i.v. + Gentamicin ** 3 mg/kg q24h i.v. HLAR (+) strains - ampicillin 2 g q4–6h i.v. + ceftriaxone 2 g q12h i.v.	HLAR (−) strains in case of immediate reaction: Vancomycin * 15–20 mg/kg based on actual body weight q8–12h i.v. + Gentamicin ** 3 mg/kg q24h i.v. HLAR (+) strains in case of immediate reaction—consultation with antibiotic therapy expert indicated
*Enterococcus faecium*	HLAR (−) strains: vancomycin * 15–20 mg/kg based on actual body weight q8–12h i.v. + gentamicin ** 3 mg/kg q24h i.v. HLAR (+) strains: consultation with antibiotic therapy expert indicated	
*Enterobacterales bacilli*	- ceftriaxone 2 g q24h, i.v. - ciprofloxacin 400 mg q8h i.v.	ESBL (+) strains: - meropenem 1 g q8h i.v. - imipenem/cilastatin 500 mg/500 mg q6h i.v.
*Pseudomonas aeruginosa*	- ceftazidime 2 g q8h i.v. - piperacillin/tazobactam 4.5 g q6h i.v.	- cefepime 2 g q8h i.v. - meropenem 1 g q8h i.v. - imipenem/cilastatin 500 mg/500 mg q6h i.v.
*Acinetobacter baumannii*	according to antibiogram, antibiotic susceptibility—difficult to predict, but usually susceptibility to: - meropenem 1 g q8h i.v. - imipenem/cilastatin 500 mg/500 mg q6h i.v. - ampicillin/sulbactam 3 g q4–6h i.v.	- colistin loading dose 9 million IU i.v. followed by 4.5 million IU q12h i.v. (at MIC = 2, increase the dose up to 12 million IU/d)
Gram(−) bacilli resistant to carbapenems	antibiotic susceptibility difficult to predict—consultation with an infectious disease specialist indicated	
*Candida* spp.	initial therapy: - liposomal amphotericin B, 3–5 mg/kg/d with or without flucytosine or - echinocandin (caspofungin 150 mg/d or anidulafungin 200 mg/d or micafungin 150 mg/d) Echinocandin may be switched to fluconazole (400–800 mg/d) in clinically stable patients with fluconazole-susceptible *Candida* spp. isolates and with negative follow-up blood culture results.	

* subsequent vancomycin doses based on concentration monitoring using AUC/MIC; ** it is necessary to monitor minimum and maximum gentamicin concentrations; *** therapy combined with gentamicin applies to patients with mediastinitis. MSSA—methicillin-susceptible Staphylococcus aureus; MRSA—methicillin-resistant *Staphylococcus aureus*; HLAR—high-level aminoglycoside resistant.

## Data Availability

Not applicable.

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
