# Peer review of "Surgical Site Infection in Cardiac Surgery"

_jcm, 2022, doi:10.3390/jcm11236991_

Round 1

Reviewer 1 Report

The article "Surgical infections in cardiovascular surgery" is undoubtedly relevant.

Among all surgical patients, the proportion of patients with infectious complications in the area of the postoperative wound is 3.5-30%. One of the common and most serious wound complications in cardiac surgery is sternomediastinitis. According to different authors, the incidence of infectious wound complications after median sternotomy ranges from 0.4% to 26%. The treatment of mediastinitis has not yet been developed definitively, and mortality in this complication remains rather high - from 3.5 to 58.3%.

The only effective and generally accepted method of treatment of patients with mediastinitis was antibiotic therapy in combination with repeated surgical intervention (wound surgery). While antibiotic therapy options remain generally accepted and their efficacy is not discussed, there have been many different surgical treatment options proposed. The basic principle is to perform resternotomy, necrectomy followed by sternum reosteosynthesis or without it and installation of permanent irrigation-aspiration, flow drainage or application of dressing with negative pressure system.

The authors have presented a literature review, considered the main trends and tendencies, described epidemiology, clinics, methods of diagnosis and treatment of SSI and DWSI. The issues of antibiotic prophylaxis and treatment are touched upon.

No comments or questions to the authors of the article.

Author Response

We appreciate the time and effort that you dedicated to providing feedback on our manuscript and we are grateful for the positive review.

Reviewer 2 Report

This is a well written review. 

This reviewer thinks the paper would be better if the authors provide more tables about prophylaxis and skin preparation as well as the antibiotics treatment.

Author Response

We appreciate the time and effort that you dedicated to providing feedback on our manuscript and we are grateful for insightful comments on and valuable improvements to our paper. We have added the suggested contents to the manuscript in the perioperative antibiotic prophylaxis section page 10 and SSI prevention section page 6. Thank you once again for the helpful comments. 

Reviewer 3 Report

I suggest that they can add photos and figures if appropriate. And little English edition.

Author Response

Thank you for this evaluative and helpful comment. All text was reviewed by medical native speaker. We have changed the text accordingly to reviewer suggestions.

The photographs and figures could be considered to add indeed, however we did not intend to use them in this manuscript, although we will certainly use them in the future. To improve the quality of the article we have added two tables to the manuscript: in the perioperative antibiotic prophylaxis section page 10 and SSI prevention section page 6. Thank you once again for the helpful comments.